# A Mixed Method Study: Defining the Core Learning Needs of Nurses Delivering Care to Children and Young People with Rheumatic Disease to Inform Paediatric Musculoskeletal Matters, a Free Online Educational Resource

**DOI:** 10.3390/children9060844

**Published:** 2022-06-07

**Authors:** Nicola Smith, Christine English, Barbara Davies, Ruth Wyllie, Helen E. Foster, Tim Rapley

**Affiliations:** 1Translational and Clinical Research Institute, Newcastle University, Newcastle Upon Tyne NE7 7XA, UK; nicola.smith@newcastle.ac.uk; 2Department of Nursing, Midwifery and Health, Northumbria University, Newcastle Upon Tyne NE7 7XA, UK; christine.english@northumbria.ac.uk; 3St. Oswald’s Hospice, Newcastle Upon Tyne NE3 1EE, UK; 4Paediatric Rheumatology, Great North Children’s Hospital, Newcastle Upon Tyne Hospitals NHS Trust, Newcastle Upon Tyne NE7 7XA, UK; ruth.wyllie@nhs.net; 5Population and Health Sciences Institute, Newcastle University, Newcastle Upon Tyne NE7 7XA, UK; h.e.foster@newcastle.ac.uk; 6Social Work, Education and Community Wellbeing, Northumbria University, Newcastle Upon Tyne NE7 7XA, UK; tim.rapley@northumbria.ac.uk

**Keywords:** education, E-learning, paediatric musculoskeletal, children and young people

## Abstract

Children and young people with rheumatic diseases and their families are often supported by nurses who may not have had specialist training in paediatric rheumatology. The purpose of our study was to establish the core learning needs of all nurses who may encounter these children and young people in their clinical practice and use this information to inform the content and format of Paediatric Musculoskeletal Matters Nursing (PMM-Nursing) Engagement with nurses working in different roles and with various levels of experience in musculoskeletal medicine informed these learning needs and PMM-Nursing content. Mixed methods ascertained learning needs under the following themes: (1) Need for increased awareness about rheumatic disease; (2) Impact of experience and nursing role; (3) Need for increased knowledge about rheumatic disease and management. In addition, our methods informed design components for an impactful learning and information resource. Representatives from stakeholder nursing groups, social sciences, and web development used this information to create a suitable framework for PMM-Nursing. The content of PMM-Nursing is now live and freely available.

## 1. Introduction

Advances in the management of paediatric and adolescent rheumatology have resulted in the emergence of highly specialist nursing roles with dependence on local support from nurses in other health contexts to enable care delivery. More children and young people (CYP) are, as a result, supported by generalist nurses who may not have experience or training in paediatric rheumatology [1]. CYP with rheumatological conditions and their families face a range of challenges daily, requiring appropriate support to maintain physical and psycho-social wellbeing [2]. These CYP need to cope with disease effects and treatment demands alongside the usual stressors associated with growing up [3]. Pain, joint stiffness, and fatigue associated with JIA can impact school performance, family life, and limit participation in physical activities. In comparison to healthy peers, children with JIA can have lower levels of physical fitness [4,5], and those with uncontrolled disease may demonstrate greater deficits when relating fitness to postural control [4]. As low levels of fitness correlate with poor psychosocial function, nurses need to appreciate that children with JIA may benefit from accessing appropriate physical fitness assessments to develop personalised fitness plans which, ultimately, have potential for quality-of-life improvement [4].

Families’ needs are complex and generalist nurses working in primary, secondary, or tertiary care may be expected to provide care, advice, and support, although their knowledge, in this field, may be relatively limited [6]. This can place nurses in a difficult situation as they will be mindful of their professional Code’s requirements [7] to: *“make sure that any information or advice given is evidence based”* [7]. To provide appropriate, up-to-date, evidence-based information, nurses need easy access to accurate specialist nursing knowledge and, despite a growing body of knowledge being available, this may prove difficult to navigate for nurses working in busy clinical roles. In addition, assurance of veracity of evidence is important to advise and support families, in line with the expectations of their Professional Code. There is, therefore, an important role for education and support of both nurse specialists and generalist nurses caring for CYP with musculoskeletal problems. 

The British Society for Paediatric and Adolescent Rheumatology Standards of Care (BSPAR) [1] for Juvenile Idiopathic Arthritis (JIA) describe the clinical service families can expect, emphasising the importance of specialist multidisciplinary teams working in clinical networks across geographical areas, delivering shared care with colleagues in primary, secondary, and tertiary care. Delivery of the standards requires a workforce with appropriate skills and training. The learning needs may vary depending on roles and responsibilities therefore educational resources must be designed with these differences in mind. 

Continuing Professional Development (CPD) in nursing is valued and integral to revalidation [8], however there seems little research evidence to guide nurses in their CPD activities. It is only recently that a competency framework has been published to provide a standardised approach for preparation for rheumatology nursing roles [9] and such education is regarded as an integral role of clinical nurse specialists [10]. Given this previous lack of a structured approach, many specialist nurses developed their expertise through being mentored and supported by expert medical staff [11]. 

Anecdotal evidence further suggests that those new to the specialist role and those working outside of the specialty continue to seek more information and guidance from senior colleagues. In addition, although nursing students are required to achieve competency in the care of CYP with chronic conditions and their families [12], they may not always gain direct clinical experience of caring for CYP with rheumatological conditions, as this may be placement dependent. It is important to first understand these differing learning needs and then design tailored education and support with these in mind. 

Access to continuing education can also be a challenge for nurses who have increasing work and home responsibilities, and innovative educational delivery methods are needed to overcome these barriers. Continuing professional development is often designed within a ‘one size fits all’ framework. This can be problematic considering that a failure to recognise different learning requirements and the context of the practice arena can change the active learner into a passive recipient of education, and thus fail to meet participant need [13]. The rapid advancement of treatments and studies in this specialism may be overwhelming for generalists or newly qualified nurses to navigate alongside time pressures in clinical practice that may act as barriers to utilising up-to-date evidence in practice [14]. Educational materials need to be accessible to those who need it at a time and place that is more convenient to them. With these concerns and the changing nature of healthcare in mind, an educational package is needed to improve access to the creators of knowledge and the nurses who use this, and in doing so meet both a professional and organisational demand for adequate and appropriate education [15]. 

Paediatric Musculoskeletal Matters (PMM—www.pmmonline.org (accessed on 31 January 2022)) is an evidence-based and peer reviewed open e-resource for paediatric musculoskeletal (MSK) medicine targeting clinicians who are not specialists in iMSK medicine [16,17,18]. PMM was launched in November 2014 and has had wide reach (221 countries with >390,153 users, >939,604 hits, Google Analytic Data October 2021) and positive feedback from around the world [19]. Through developing PMM-Nursing (www.pmmonline.org/nurse (accessed on 31 January 2022)), we aim to build on the success of PMM and expand the reach and impact to include nursing groups. 

This study aimed to establish the core learning needs of nurses working in different roles across primary, secondary, and tertiary care, and those delivering care to CYP with rheumatic disease. This information was then used to develop the content for PMM-Nursing. 

## 2. Methods

Engagement with potential users of the educational resource informed the core MSK learning needs to derive content addressing the needs of nurses working in different roles to deliver care to CYP.

A criterion sample of qualified and student nursing professionals were invited, via their links to professional networks and higher education institutions, to participate. Participants were included if their professional role could involve caring for CYP with rheumatological disorders and their families, or early detection of signs of disease in CYP. 

Mixed-method research was conducted using an online survey of expert rheumatological Clinical Nurse Specialists (CNS) within the BSPAR (CNS, *n* = 27). This survey (response rate: 77%, with 27/35 respondents) ascertained expert opinion about educational needs based on experience, knowledge, and skills, and how these mapped to different nursing groups. Survey results informed the questions for the focus groups exploring educational needs of nurses working outside the specialty: nursing students (*n* = 15), health visitors (*n* = 3), and nurses from general paediatrics (*n* = 2), community (*n* = 7), school health (*n* = 3), research (*n* = 5), and adult rheumatology (*n* = 4). Focus groups were recorded and transcribed and focused on the required content and format for provision. 

The survey data was analysed using descriptive statistics, but where free text comments were included these responses were analysed using qualitative techniques. Focus group transcripts were analysed qualitatively following standard procedures for qualitative analysis, including open and focused coding, constant comparison, and deviant case analysis [20]. Finally, a consensus group was held to refine and further develop the core learning needs. From this, a final list of learning needs emerged which representatives from stakeholder groups, social scientists, and web development experts then transformed into a suitable framework for PMM-Nursing (see Figure 1). 

This study had ethical approval from Newcastle University’s Faculty of Medical Sciences Ethics Committee (00711/2013), with informed consent from all participants. 

## 3. Results

Centrally, the CNS survey results informed the questions within subsequent focus group discussions with other nursing groups. As such, the presentation of these results represent an outcome of this iterative process and focus on the final stage, where ideas around form and content of the educational resource were established. 

Data analysis revealed the following themes: (1) Need for Increased Awareness about Rheumatic Disease, (2) Impact of Experience and Nursing Role, (3) Need for Increased Knowledge about Rheumatic Disease and Management. In addition, participants identified key elements of the format and content required for an impactful learning and information resource. These findings guided the design and content of PMM-Nursing (see Table 1 for a breakdown of knowledge required by nursing group). 

### 3.1. Core MSK Learning Needs


(1)Need for Increased Awareness about Rheumatic Disease


Those without professional experience of paediatric rheumatology expressed a need for increased awareness about the presentation of rheumatic disease in CYP. This started from a very basic level, focusing on the existence of rheumatic disease in CYP. For example, a Student Nurse outlined that, ‘It’s a condition really that you’d associate with older people … people wouldn’t even imagine they could have such a thing’. Further comments indicated a lack of awareness of other fundamental issues including age of onset and presentation of signs and symptoms. It was apparent that for the health visitor group particularly, knowing ‘what age group to even consider problems with rheumatology’ was something they were unaware of, yet this is a core piece of information that would enable them to identify CYP with suspected rheumatic disease within their everyday practice and provide appropriate support and reassurance. 

The Health Visitors acknowledged that key information should provide guidance on ‘what is normal MSK development’, ‘when to be concerned’, and ‘what to do when concerned’ to empower professionals outside the specialty to refer effectively and appropriately.

The CNSs also identified a lack of awareness amongst many nurses outside of the paediatric rheumatology specialism, as a key area of concern that needs addressed. 

One of the most important aspects of care is that the nursing staff, looking after the child, in whatever setting, are able to recognise/identify if the child has a problem and know where to seek the appropriate assistance to deal with this.(CNS)

They outlined that a lack of awareness that rheumatic disease exists in CYP, alongside poor knowledge, can result in it being challenging to ensure a child’s needs are met and that increasing awareness could contribute to reducing ‘delays in access to care’ (CNS).


(2)Impact of Experience and Nursing Role


Learning needs were dependent on past experience and expectations within their current nursing role. For example, health visitors focused on normal development and the support they could offer to parents, whereas school nurses focused on the information they would require to support the CYP within the school setting.

I’d probably want some basic information about specific conditions that’s easy to understand because we’re often expected to explain to school staff as well, and then the impact that that condition has on a school day … For us it would be helpful to explain what the condition is and then how that then impacts on their day-to-day life. (School Nurse)

Similarly, adult nurses focused their learning needs on the key disease groups they would expect to encounter. For example, one nurse outlined that as they only see a specific group in their transition clinic—connective tissue patients—they would not want to ‘give myself more work’, by focusing on understanding that group in any real detail.

In contrast, the research nurses and general paediatric nurses highlighted that their knowledge needs can differ dependent on either the study they are involved with or the patients they are caring for at that specific time. 

Rather than a generalist, we need to know everything about rheumatology. We actually need to know that pocket [of information] that relates to the study that we’re doing at that time....(Research Nurse)

In this way, learning needs are closely situated to ongoing responsibilities and so are often emergent and ever changing depending on the nurse’s role at that time. 

I think like a sort of easy look database… so when somebody comes on you go ‘oh I’ve not looked after one of them before’ and then you can have a look and say ‘oh that’s what it is’.(General Paediatric Nurse)

For those nursing groups with more emergent and situated needs, it is important to enable them to tailor their own learning, depending on the knowledge needed for their individual roles. 

Unsurprisingly, those with professional experience and greater knowledge of rheumatic disease were more able to define their learning needs, whereas those with no practical experience offered a more general discussion of their potential needs. For example, the community nurses, a group experienced in managing children with rheumatic disease commented: 

Basically for us, as a team, I don’t know about the other teams, but we basically just go out and give the injections [to patients with JIA] and just a bit of support for the families really. So, knowing more around like you say, the side effects and the ongoing effects that might be pertinent to the children would be interesting so we can support them in the right direction.(Community Nurse)

As such, those with practical experience focused on what would be required within their nursing role, and any areas of knowledge or support that could enable them to deliver their role more effectively.

The nursing students acknowledged that they were unsure what nursing role they might choose once qualified. As a result of this they felt a complete overview of what could be expected within the different nursing roles as well as what would be required for any future placements would be useful. When asked to identify what they think these areas of knowledge might entail they considered past experiences—either personal or from placements—with similar considerations: 

My grandma has rheumatoid arthritis so I’ve kind of seen what it does to old people but I wouldn’t know how it would affect a child… It’ll be useful to have that comparison. (Student Nurse)

In the focus group students worked through ideas exploring a case-based scenario. For example, highlighting that ‘if they are looked after in the community it just might be their day to day stuff’ that they would consider where as if ‘they’re in hospital it’s obviously going to be more,’ suggesting a more in-depth assessment and input may then be required. With this in mind, there is a need for tailored education to meet ever changing and individual role dependent knowledge requirements, to ensure practice is evidence based. 


(3)Need for Increased Knowledge about Rheumatic Disease and Management


The need for increased knowledge was highlighted across all the nursing groups. Key topics included (1) Disease knowledge, (2) Management, (3) Impact of disease/condition on CYP and their family, (4) Available support for the nurses, CYP and their families, and (5) Current and topical research—see Table 1 for a breakdown of knowledge required by nursing group. 

Whilst the detail of the knowledge needed differed across the nursing groups depending on what would be expected within their role, the key areas nevertheless tended to comprise of the same topic areas. For example, whilst the health visitors acknowledged management of treatment is something they ‘are not going to be dealing with directly’ but rather something that is ‘good to know… so you can be watching, and signposting,’ the paediatric nurses needs required more specific information relating to particular treatment regimens:

Who they decide needs the steroids … the observations required so like a 2 h infusion and half hourly obs. and then an hour afterwards … how long [they] stay for ‘cos they are different to gastro patients.(General Paediatric Nurses)

All the nursing groups asked for information that provided a basic understanding of rheumatological conditions affecting CYP; including signs and symptoms of the conditions and disease processes in the first instance, but some required additional information they deemed relevant for their role. For instance, the adult nurses asked for this disease knowledge section to provide a comparison to adult conditions, guidance on the impact of age on the disease process, and to highlight the conditions that affect patients health the most; both the research nurses and student nurses asked for this section to cover anatomy and physiology linked to the conditions and what is affected, and the health visitors asked for guidance focusing on the conditions in relation to normal development—when are things normal vs. when there is a need for concern or further investigation is warranted. 

In a similar manner, all the nursing groups asked for a basic understanding of the medications used to treat these conditions and details on why they were being used. This was a starting point for most of the nursing groups, providing a base on which to build their understanding further to include more detailed treatment specific information including when medication can be given (and not), when to withdraw treatment, how to administer safely, and what the side effects are; for others, such as the health visitors, a basic understanding was sufficient. The adult rheumatology, general paediatric, and student nurses were further interested in guidance on typical treatment plans. For the adult rheumatology nurses, there was particular concern with how management differs between paediatric and adult care, and they sought specific detail on transition processes. The general paediatric and student nurses sought similar comparisons, but rather than simply focusing on comparisons between child-adult services they focused on how treatment differs from that of other conditions e.g., differences in dosage (paediatric nurses).

Some of the nursing groups (adult rheumatology, general paediatric, school, and student), asked for information concerning the immunosuppressive nature of the treatments, the impact of infectious disease, and how these impact vaccination, to enable them to inform parents and put relevant steps in place to protect the child. Information concerning the role of the MDT and where available support could be found was seen to be a key area, particularly across the nursing groups, for which support from other colleagues was seen as central within their role (e.g., general paediatric, school and student). 

Advice on the impact of the conditions was seen as important across all the nursing groups except the general paediatric nurses; whereas for the community, health visitors, and school nursing groups this focused on the support they could provide, other nursing groups (adult rheumatology and student nurses) instead focused on support they could signpost patients to. Both the student nurses and adult nurses acknowledged the need to encourage normality and asked for links to anything to support this, whereas the school and community nurses instead focused on the care they would be giving and any guidance that would help within the role. For example, for the school nurses, an increased understanding of the impact on the school day and guidance on any activities the child should avoid or that require additional support, and what support (if any) should be put in place within the school setting, were seen as advantageous. In a similar way, the community nurses considered what support would likely be expected within their role and sought further information on the full range of symptoms rheumatology patients may experience and how they could help, quoting typical questions such as ‘Why do I feel sick?’, ‘Is it normal to be tired?’, ‘Why do you get the red lump? How long you can expect it to be there? What’s acceptable and what’s not acceptable?’. 

The community, adult rheumatologists, and school and student nurses were aware of the key role signposting plays within their everyday practice, and sought detailed information on support available to both the patients and themselves. 

Key areas within their role were highlighted and these included safeguarding (adult rheumatology and school nurses), career/employment/education and sexual health advice (adult rheumatology nurses), and chicken pox contact (school nurses). Finally, all nursing groups excluding school nurses expressed interest in more information surrounding current, relevant research. Research nurses focused on National Institute for Health and care Excellence (NICE) guidelines, whereas other nursing groups had broader interest in new research, treatments, and developments, to enable ‘up to date’ practice. All nurses acknowledged that any treatment or guidance they provide should be ‘research and evidence based’. They wanted to ‘know what the current information is’ to feel well equipped to respond to the information needs of families acknowledging that there can be ‘a high proportion of really educated families’ that require nurses to ‘know what the current information is’. As such, any information or opportunity to enable them to update was welcomed. 

The CNS’ identified the same topic areas as other nursing groups as being key to knowledge development for their role. Recognising the varying levels of experience of CNSs, there was a desire for information to be more in-depth to guide development to the expert level. Whilst these five topics areas were important across most of the nursing groups, albeit to different degrees, there were some exceptions. For example, current and topical research appeared unimportant for the school nurse group, as their discussions focused on knowledge enabling holistic support for children in school settings. The general paediatric nurses viewed the impact of disease and treatment as being managed by other members of the MDT:

That (impact) again isn’t us … That’s more, the physios come up and down to see the patients when they are here … and the specialist nurses would come. (General Paediatric Nurses)

Similarly, general paediatric and research nurses saw their role in supporting the child and family short term. 

I would be making sure I told their specialist nurse about what my concerns were. Because they are much better placed…to make sure they get the support and the right advice that they’re going to need to manage. (Research Nurses)


(4)Design Components for an Impactful Learning and Information Resource


Design components required for creating an impactful resource were provided. These listed details on the preferred format, structure, and layout to create an educational resource suited to nurses. See Table 2 for a detailed breakdown of the identified six main areas. 

The depth and breadth of information presented needs to reflect the different educational needs of nurses working in a variety of roles. Non-paediatric rheumatology nursing groups acknowledged that they are not specialists within the area and resources should be pitched in a manner that reflects this. Equally, CNSs recognised that other nursing groups are not specialists in rheumatology, nor should they be expected to be.

It would be difficult for nurses working in day care, community, or clinical settings to undertake in-depth education on all of the specialties who may use their areas. Some basic knowledge of diseases, medications, and side effects, which is easily accessible to them, would be of benefit. Learning can take place with further support from the paediatric rheumatology team specific to the child, who will have more up-to-date knowledge of the child, their family, and their needs in regard to their illness. One of the most important aspects of care is that the nursing staff looking after the child in whatever setting are able to recognise/identify if the child has a problem and know where to seek the appropriate assistance to deal with it. 

It was further acknowledged that while the site does not need to identify content specific to each nursing group, it should adopt a novice to expert approach and enable nurses to access the information they need to match their role and interests.

Be nice to have access to see what the school nurses would look at. Because obviously if a school nurse could ring us up and say ‘Tommy Jones is coming into school and he’s tired and he aching’ and all the rest of it. ‘Is that normal and what can he do and what can’t he do?’ So that would be nice’.(Community Nurse)

Feedback also identified that the presentation of content could enable more effective learning through the adoption of a case study approach. 

It makes it more real though, doesn’t it? If you can link into something. It stops being something that somebody somewhere might have and it becomes what real people have.(Student Nurses)

Providing users with a real-life picture of what the diseases look like from both a nursing and patient/family perspective enables users to apply their knowledge to work through example case scenarios and helps promote active learning, critical thinking, and problem solving, all of which are important features of nursing practice [21].

### 3.2. Peer Review and Initial User Feedback

Peer review of the website was undertaken by nurses working in areas matching the target audience (*n* = 10) and confirmed suitability of both content and design of PMM-Nursing. Suggestions for additional content or amplification of details were welcomed and addressed in the content. Feedback was very positive, with PMM-nursing viewed as an informative and credible resource for a wide target audience across the nursing profession. 

Works for all levels of understanding, for students it gives them an introduction of what conditions to expect and for trained staff allows them to build on their knowledge and seek out anything further.(Ward sister)

Ongoing Royal College of Nursing (RCN) endorsement adds a further layer of content validity and, furthermore, peer review and feedback rated PMM-Nursing as having an impact on their clinical practice. The site was seen as engaging, providing useful nursing guidance to aid understanding, identification, and management of rheumatological conditions. 

## 4. Discussion

There is a wide spectrum of nurses across hospital and community settings who care for and support CYP with rheumatological conditions. The learning needs of these nurses vary depending upon roles, responsibilities, and experience. 

This study has identified core learning needs for all nursing groups and developed an educational resource–PMM-Nursing (www.pmmonline.org/nurse (accessed on 31 January 2022)). Adoption of a novice to expert approach [22] in the organisation and presentation of educational material enables nurses at all stages of their learning trajectory to access knowledge appropriate to their needs. In contrast, identification of learning needs within other specialisms has led to the emergence of a range of education, competency, and career framework documents, e.g., in cancer nursing [23], diabetes nursing [24], and neonatal nursing [25]. PMM-Nursing, however, is more aligned with NHS Health Education England’s e-learning offer, where there has been e-resource development addressing specific learning needs, e.g., in safeguarding, patient safety, suicide and self-harm, and end-of-life care [26]. As with PMM-Nursing, the development of these e-resources, as a means of continued professional development, anticipates improvements in knowledge leading to enhanced care delivery. 

The PMM-Nursing content is presented in six modules entitled: *Normal Child; Arthritis and Conditions; Assessment; Management; Cases for Discussion; Resources*. Within each module, sub-modules present the information in complementary modes (videos, images, Top Tips, cases, summary points and links to recommended websites, guidelines, and key references). Levels of learning informed the content development and information is structured to address the learning needs of nurses working in different roles (see Figure 2).

Content for the website was written by the project team with a structured content approval process and final ‘sign off’ by senior members of the clinical team. PMM-Nursing is complementary to PMM and addresses the different needs for the wider nursing community. A process of external peer review further assessed the content, format, and user experience, and confirmed the suitability of PMM Nursing in its current form. Initial informal user feedback is positive and a process of ongoing development in response to future user feedback is planned. From a learning and teaching perspective, website content is embedded locally within preregistration nursing programmes as part of preparation for formative and summative assessments.

Through the use of social media and twitter, embedded within the website, the aim is to establish a continued presence and to stay connected with the those using the site. By engaging with our audience, we will be continuously promoting the site whilst building our reputation and raising awareness. 

### 4.1. Limitations

The content and design of PMM-Nursing was guided by findings from consultation with potential users of the resource and although the expert group of nurses represented services across the UK, other groups were solely recruited from the North of England, and so views expressed may not reflect the needs of nurses elsewhere. Likewise, there were limited representatives recruited from general paediatrics and health visitor settings. However, subsequent user feedback and peer review via the RCN endorsement process has indicated that PMM-Nursing does meet the learning needs of these groups and has been well received by users.

Although the learning needs of nurses was the focus of this study, future developments would benefit from the inclusion of CYPs views of care experiences and their preferences in the way their care and support is delivered. 

### 4.2. Future Implications

It is likely that specialist rheumatology teams will continue to depend on nurses in different healthcare contexts to provide on-going care and support for CYP with rheumatological conditions and their families. PMM-Nursing has the potential to address the unmet learning and it is envisaged that nurses using PMM-Nursing will be better equipped to deliver effective care and support, the outcomes of which need further exploration from a CYP perspective. The novice to expert approach provides education and guidance to support those working in varying roles. Our aim is to have ongoing endorsements from the RCN and to seek similar endorsements from similar professional organisations for nurses around the world. Our aspiration is for PMM-Nursing to be the ‘go-to’ online resource to support nurses with different professional experiences and roles, and to support the evolution of the specialist CNS role internationally. The role of specialist nurses as integral members of the MDT has yet to develop in many parts of the world, and is deemed to be a barrier to clinical care delivery, especially in low- and middle-resource settings; hence, workforce capacity building is a priority theme for the Paediatric Global MSK Task Force [27]. 

## 5. Conclusions

Our collaborative and evidence-based approach, with the engagement of end users and stakeholders, aims to ensure that PMM-Nursing addresses the core learning needs of all nurses working with CYP with rheumatological disease and their families. PMM-Nursing is now live, free, and available on mobile, tablet, or PC. Evaluation is ongoing through Google Analytics and in-resource e-surveys, with future work planned to evaluate PMM-Nursing further in terms of user experience and clinical impact.

## Figures and Tables

**Figure 1 children-09-00844-f001:**
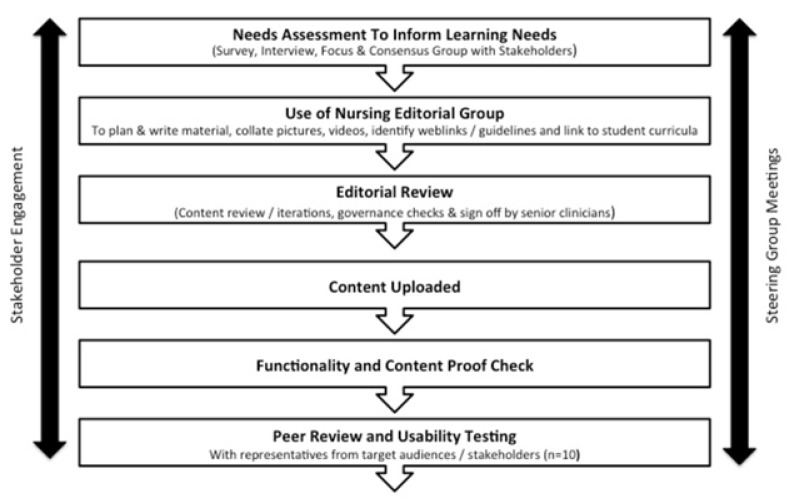
PMM Nursing Development Process.

**Figure 2 children-09-00844-f002:**
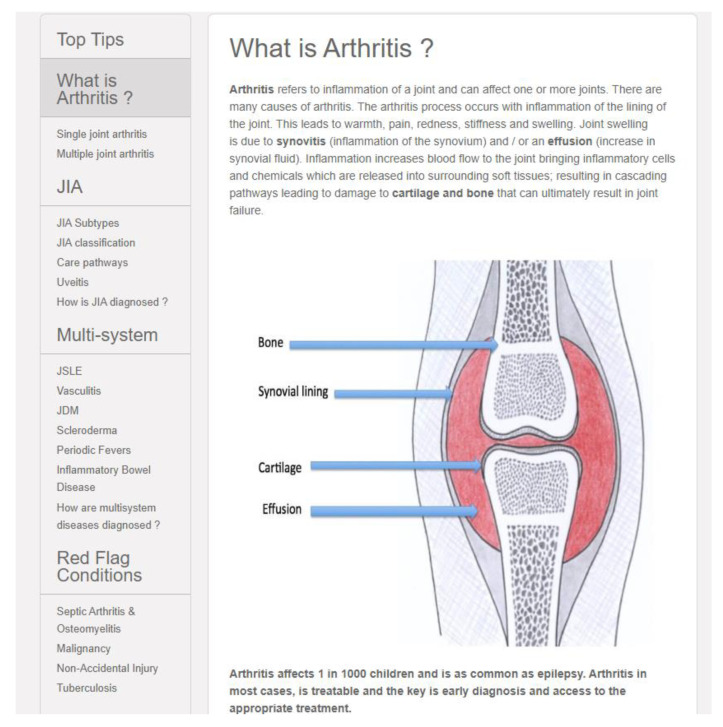
Example of novice to expert design.

**Table 1 children-09-00844-t001:** Breakdown of knowledge required by nursing group.

	Disease Knowledge	Management	Impact of Sisease/Condition	Available Support	Current and Topical Research
Paediatric rheumatology nurse specialist	Understanding of all rheumatological conditions affecting children and young people including the signs and symptoms of the conditions and disease processes, diagnosis process, classifications, how to manage these conditions and the investigations needed and why. Recognition of how these are different to adult conditions. Anatomy, physiology and pathophysiology– particularly that of joints and the musculoskeletal system. Blood test monitoring, regimes, results and what to do for abnormal results.	Common facts, tips and advice. Examination techniques including joint examinations/assessment and clinical examination skills including communication, key questioning, PGALS and diagnostic reasoning. Knowledge of how to take a clinical history. Knowledge of medications used to treat these conditions, why they are being used, when they can be given or not, when to withdraw treatment, how to administer safely, what the side effects are (and how to deal with them – including immunosuppressive nature of treatments) and monitoring. Safe handling of chemotherapy preparations and knowledge that Methotrexate is not a poisonous drug. Guidance on giving injectable therapies in particular to challenging patients and how to safely administer Subcutaneous and intravenous medication e.g., DMARD’s and biologics. Guidance on pain management and self-management of the condition. Prognosis of long-term conditions. What to expect in adult care/enabling adolescents to prepare for transition to adult service. Guidance on goals of treatment - Wallace criteria. Knowledge of roles and value of MDT within paediatric rheumatology and how to coordinate patient care. Assessment of disease activity, wellbeing and adjustment to illness.	An understanding of what it is like living with chronic disease/knowledge of the patients’ journey in relation to the most common rheumatological illnesses. Impact on the child and family and how to provide support. Effect on education and employment.	Up to date knowledge of safeguarding. Lifestyle and sexual health advice. How to tips e.g., letter writing to GP and other health professionals. Knowledge to educate patients, families and other professionals. Awareness of other support services and self help groups/charities to sign post families	National picture of paediatric rheumatology/National recommendations. Knowledge regarding ongoing research. Knowledge of national groups working to improve care and share information.
Adult rheumatology nurse	Basic understanding of rheumatological conditions affecting children and young people (in particular JIA and the subtypes of JIA), including the signs and symptoms of the conditions and disease processes, how to manage these conditions and the investigations needed and why, and how these are different to adult conditions. Highlight which conditions/subtypes of JIA are the main ones that affect patient’s health the most. The impact of age on the disease process.	Basic knowledge of the medications used to treat these conditions, why they are being used and why treatment is different for children and adults; including what age children can start biologics. Typical treatment plans for each condition. Detail on how children are typically managed highlighting areas where this differs to typical adult care. Differences between adult and child clinic set up. Detail on the transition process - increased understanding what paediatric team are doing before child comes to adult service and information to help them understand what the patient typically experiences in paediatrics. Immunisation advice.	Impact on child and family and where to refer patient/family to for information or support. The need to encourage normality and links to anything to support this.	Who to contact for advice/information/where they can go to obtain relevant information. What to do/where to go if they have safeguarding concerns. Information on career/employment/education support available and where to direct patients/families for this. Sexual health advice. Any relevant information that they can direct patients to.	What’s new in the area. New research. Recent guidelines. Topical articles.
Community nurse	Basic understanding of rheumatological conditions affecting children and young people including the signs and symptoms of the conditions and disease processes, and how to manage these conditions.	Basic knowledge of the medications used to treat these conditions, why they are being used, when they can be given (and not), how to administer safely and what the side effects are. Detail on the ongoing effects of treatment, dosage effects and effect of medication changes. What can be expected, how soon they will see/feel the effects etc. Guidance on how to recognise when have symptom control. Guidance on treatment follow up – e.g., how often do they need bloods etc. Home delivery details.	Impact on child and family and how to provide support. Provide full range of symptoms and guidance e.g., why red lump and what can be done? Why feel sick? What painkillers can the patient take? Advice for other professionals about what child can and cant do and what child should be like if treatment under control.	Who to contact for advice/information/where they can go to obtain relevant information. Support to direct parents/patients to.	New research. Links to useful articles. Highlight new treatments.
General paediatric nurse	Basic understanding of rheumatological conditions affecting children and young people including the signs and symptoms of the conditions and disease processes, and the investigations needed and why.	Basic knowledge of the medications used to treat these conditions, why they are being used, when they can be given (and not), when to withdraw treatment, how to administer safely and what the side effects are. How medications and treatment differs to that for other conditions e.g., differences in dosage. Basic blood monitoring requirements. Highlight where they differ to other conditions for example different blood tests, urine tests, observations etc. The process of treatment – whether need labs etc. Example of ‘normal’ patient experience – typical treatment plan. What to do in case of viral infections for patients on treatment and the need to recognise the immunosuppressive nature of treatments. Role of MDT and its importance and how rheumatology care is organized – when to speak to someone else and who to speak to. What to do if something goes wrong.	N/A for their role.	N/A for their role.	What’s new in the area. New research. Links to protocols/drug protocols. Highlight new treatments.
Health visitor	Basic understanding of rheumatological conditions affecting children and young people, including the signs and symptoms of the conditions (how likely to present) and what to look out for. Normal development (when are things normal and when aren’t they).	Basic knowledge of the medications used to treat these conditions. Good to know so they can signpost but not going to deal with directly. Guidance on what would ease symptoms.	Impact on child and family and what support they could/would offer in their role.	Support for parents – parent information sheets.	New research/current studies that may be relevant. Guidelines.
Research nurse	Basic understanding of rheumatological conditions affecting children and young people, including the signs and symptoms of the conditions and disease processes, and how to manage these conditions. Anatomy and physiology linked to the conditions and what’s affected.	Basic knowledge of the medications used to treat these conditions, why they are being used), what the side effects are and when to raise concern. Would be guided if using them so no in depth knowledge needed. Guidance on explaining things to children.	Impact on child and family. Quality of life. Coping with nausea.	N/A for their role.	NICE guidelines.
School nurse	Basic understanding of rheumatological conditions affecting children and young people, including the signs and symptoms of the conditions and how to manage them.	Basic knowledge of the medications used to treat these conditions (names so to recognise them), when they can be given (and not), how to administer safely and what the side effects are. Guidance on interaction with other things. The need to recognise the immunosuppressive nature of treatments and the impact of infectious disease. Role of MDT – who on team to contact and when. Emergency care and when to refer and to whom. Guidance on accidents e.g., what to do if they fall, will they have a bleed etc. Immunisation guidance. Typical care plans.	Impact on child and family, and guidance how they can provide support. Impact on school day and what support (if any) needs to be put in place at school. Information on expected school attendance and any impact that condition may have on this e.g., exceptions for late start. Advice on exercise and activities they should avoid (if any).	Who to contact for advice/information and a link to member of staff where possible. Chicken pox contact for patients on biologics, steroids and MTX and guidance on what they do. Support for school staff/information sheets. Guidance on where to signpost others too. Guidance on support/services for young people that they can signpost patients too. Safeguarding issues.	N/A for their role.
Student nurse	Basic understanding of rheumatological conditions affecting children and young people, including the signs and symptoms of the conditions and disease processes, and how to manage these conditions. Anatomy and physiology linked to the conditions and what’s affected.	Basic knowledge of the medications used to treat these conditions, why they are being used, when they can be given (and not), when to withdraw, how to administer safely and what the side effects are. Guidance on interaction with other things. Detail on how children are typically managed. Guidance on how to recognise when have symptom control. How treatment would differ depending if treatment is within a hospital or community setting. Information specifying differences that may exist in relation to management across countries or counties. What to do in case of viral infections for patients on treatment. The need to recognise the immunosuppressive nature of treatments and the impact of infectious diseases. Role of the MDT and its importance, and how rheumatology care is organized – who to speak to and when.	Impact on child and family and where to refer patient/family to for information or support. The need to encourage normality and links to anything to support this. Impact on school day and what support (if any) needs to be put in place at school. Information on expected school attendance and any impact that condition may have on this. Advice on exercise and activities (if any) they should avoid. Quality of life.	Support to direct patients/parents to. Guidance on support/services for young people they can signpost patients too. Parent information sheets. Support for school staff/information sheets. Guidance on where to signpost others to. Any relevant information that they can direct patients to.	What’s new in the area. Topical/useful articles. Highlight new treatments.

**Table 2 children-09-00844-t002:** Design Components to Consider for an Impactful Learning and Information Resource.

Subtheme 1: Pitch
Design to reflect non-specialist nature of most user groups. Fairly simple language, designed with a layperson in mind and free from jargon. Practical knowledge reflecting each nursing groups educational needs to be presented in a manner that allows users to identify and choose which aspects they wish to look at.
Subtheme 2: Accessibility
Resource should be compatible across all platforms and a mobile compatible version should be designed in addition to a web-based version Resource should also be compatible with older operating systems (that are typical of the National Health Service (NHS) workplace) and work when Internet access is limited wherever possible. Site navigation using as few clicks as possible: bookmarking facility allowing further exploration when time allows, for example, at home, at work or whilst travelling to patients’ houses.
Subtheme 3: Content presentation style
Resource design should be colourful and interesting to attract and keep user attention Style should not focus heavily on text-based presentations but instead should utilise varied mediums to prevent the user becoming ‘bogged down’ with text enabling a more interactive experience. For example, using flowcharts, diagrams, pictures, photos, videos, quizzes, question-answer presentations and case histories. Text should be presented in sections, starting with a concise statement or brief summary of key points before moving to more detailed information Top tips or key summary documents should be included to gain a brief overview of a topic area in the first instance or as a refresher resource at a later date.
Subtheme 4: Format
Format should break content into relevant sections and include a specific resources section incorporating patient resources, printable information sheets, guidelines and recent and topical articles. A simple resource layout with a clear menu bar and tabs highlighting how a user can navigate the resource. An effective facility enabling keyword searches.
Subtheme 5: Credibility
Credibility should be enhanced through: - The use of institution logos (University/hospital). - The inclusion of evidence-based content. - References and links to information sources. - A list of professionals contributing to resource content. - Exclusion of pop-ups or adverts. - Professionally presented content with correct spelling and grammar.
Subtheme 6: Governance of sensitive content
Email login to access some of the more sensitive content (videos) ensuring governance compliance and providing reassurance that the resource is credible and responsible. Acceptability further improved by providing the opportunity for users to effectively organise information for their own use (bookmarking pages, a personal dashboard for ‘favourite/most useful’ pages or to collate a log of accessed pages as evidence of access to the resource). Whilst some users may be reluctant to create ‘another password to remember’ being able to access some content without a password (non-sensitive materials) would be beneficial. Where a password is needed the preferred option is to have a personal password rather than one which is computer generated.

## Data Availability

The data presented in this study are available on request from the corresponding author.

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
