# Peer review of "A Mixed Method Study: Defining the Core Learning Needs of Nurses Delivering Care to Children and Young People with Rheumatic Disease to Inform Paediatric Musculoskeletal Matters, a Free Online Educational Resource"

_children, 2022, doi:10.3390/children9060844_

Round 1

Reviewer 1 Report

This is a nice work that addressed the core learning needs of all nurses working with children with rheumatic diseases and MSK disorders and their families. It will be an informative on-line resource to support the concerned nurses.

No further comments.

Reviewer 2 Report

This is a very well written manuscript regarding the very important topic of establishing the supportive role of a nurse as a member of the professional team which is taking care of CYP. However, I must stress I am slightly disappointed with the impact of the results and very general conclusions without a clear take-home message. The diagnosis made within this research is quite clear for a potential reader even without reading it closely - but what is the cure? I suggest expanding conclusions with some insightful idea e.g. encouraging nursing students to share the basic info about PMM-Nursing via social media. We know already what has to be done - I am interested in the authors' concept how to do it.

One important limitation of the study which might have been overlooked is consulting CYP about their expectations on nurses' support. It would provide the relevant direction of future endeavour of PMM. I suggest the authors write an extra paragraph about this issue.

My version of the manuscript did not include Figure 1. However, it did contain Figure 2 which is hardly informative and I would transfer the list of topics into the main text. 

It is my belief that the Introduction would benefit from an extra Figure illustrating the potential impact on CYP with being taken care of by unspecialised personnel.

Reviewer 3 Report

Thank you for the opportunity to review this study. In general, the paper is well written, but there are some drawbacks.

  • Title: The study design is unclear. Insert design both in the title and in the manuscript
  • Introduction: The assessment of fitness levels and motor practice is not included. Juvenile idiopathic arthritis is the main cause of physical disability. George Frederic Still showed the destruction of cartilage and the joint deformity related to tissue contractures caused by a lack of joint mobility. Pain and fatigue are common in subjects with JIA and can influence school performance, family life, and an inactive lifestyle. Psychosocial functioning is correlated with low levels of fitness. Assessment of fitness levels are important to these individuals. I would integrate the introduction with: Patti, A.; Maggio, M.C.; Corsello, G.; Messina, G.; Iovane, A.; Palma, A. Evaluation of Fitness and the Balance Levels of Children with a Diagnosis of Juvenile Idiopathic Arthritis: A Pilot Study. Int. J. Environ. Res. Public Health 2017, 14, 806. https://doi.org/10.3390/ijerph14070806
  • The purpose of the study is unclear
  • There is no in-depth description of the sample

Results

  • In the manuscript, reference is made to “Data analysis” but the analysis is not present
  • Multi-method research was conducted using a survey, focus groups and interviews with clinical nurse specialists (CNS, n=28), nursing students (n=15), health visitors (n=3) and nurses from general pediatrics (n=2), community (n=7), school health (n=3), research (n=5) and adult rheumatology (n=4)”. Have they all been analyzed together or separately?
  • I would suggest rewriting the results section. A quantitative analysis of the responses of the whole sample (rather than inserting examples of responses) would give clarity and strength to the study

Round 2

Reviewer 3 Report

I thank the authors for following some of my suggestions. It is my opinion that the purpose of this study is clearer for the reader

However, including reinforcement on the fitness level and quality of life of children with rheumatoid arthritis in the introduction could be significant. In 2017, Kwon, H. J. et suggested that children with juvenile rheumatoid arthritis (JRA) have inferior physical fitness when compared to healthy children. Furthermore, in 2021, Sobejana, M et al showed that the subjects with rheumatoid arthritis at an increased CVD risk, associated with a low level of physical activity. 

For nurses, making them aware that a fitness test battery is easily administered could be important.

I suggest integrating the introduction with: Patti, A.; Maggio, M.C.; Corsello, G.; Messina, G.; Iovane, A.; Palma, A. Evaluation of Fitness and the Balance Levels of Children with a Diagnosis of Juvenile Idiopathic Arthritis: A Pilot Study. Int. J. Environ. Res. Public Health 2017, 14, 806. https://doi.org/10.3390/ijerph14070806

Author Response

We have responded to your suggestions and inserted text into the introductory paragraph. References added to reference list.  Changes highlighted in yellow. We trust this will address the issues you have highlighted to us. Many thanks. 
